# No Glycation Required: Interference of Casein in AGE Receptor Binding Tests

**DOI:** 10.3390/foods10081836

**Published:** 2021-08-09

**Authors:** Hannah E. Zenker, Malgorzata Teodorowicz, Harry J. Wichers, Kasper A. Hettinga

**Affiliations:** 1Food Quality & Design Group, Wageningen University & Research Centre, 6708 WG Wageningen, The Netherlands; hannah.zenker@wur.nl; 2Cell Biology & Immunology, Wageningen University & Research Centre, 6700 AH Wageningen, The Netherlands; gosia.teodorowicz@wur.nl; 3Wageningen Food & Biobased Research, Wageningen University & Research Centre, 6708 WG Wageningen, The Netherlands; harry.wichers@wur.nl

**Keywords:** casein, sRAGE, CD36, immunogenicity, milk protein

## Abstract

For the determination of the binding of heated cow’s milk whey proteins such as β-lactoglobulin to the receptors expressed on immune cells, inhibition ELISA with the soluble form of the receptor for advanced glycation end products (sRAGE) and scavenger receptor class B (CD36) has been successfully used in the past. However, binding to heated and glycated caseins in this read-out system has not been tested. In this study, inhibition ELISA was applied to measure the binding of cow’s milk casein alone, as well as all milk proteins together, which underwent differential heat treatment, to sRAGE and CD36, and we compared those results to a dot blot read out. Moreover, binding to sRAGE and CD36 of differentially heated milk protein was measured before and after in vitro digestion. Casein showed binding to sRAGE and CD36, independent from the heat treatment, in ELISA, while the dot blot showed only binding to high-temperature-heated milk protein, indicating that the binding is not related to processing but to the physicochemical characteristics of the casein. This binding decreased after passage of casein through the intestinal phase.

## 1. Introduction

Immunogenicity and allergenicity of milk protein (MP) can be altered by heating [1]. Next to changes in hydrophobicity, negative charge, and protein aggregation, advanced glycation end products (AGEs) have been identified as a factor that modulate immunogenicity [2,3,4,5,6,7,8]. The receptors for AGEs have an important role in the recognition of AGEs by innate immune cells. However, as pattern recognition receptors, they are also able to bind to other structures, such as lipopolysaccharide, members of the S100 protein family, high mobility group protein box-1, amyloid-β, and fibrillar protein aggregates [9]. Recent studies reported the binding of the soluble form of the receptor for AGEs (sRAGE) to aggregates of the heated MP β-lactoglobulin [2,10]. However, mammalian milk contains two main protein fractions: casein and whey proteins, of which the caseins are the most abundant (80% of total protein in bovine milk), whereas whey proteins are reported to be the main allergen in milk. Although whey protein has been extensively studied in relation to heating, binding of sRAGE towards heated caseins has not been investigated. In contrast to the globular whey proteins, caseins are proteins with minimal secondary structure that assemble in micelles with a mean diameter of 200 nm [11]. Although casein micelles are stable upon heating, they show temperature- and pH-dependent dissociation of αs- and β-caseins below 100 °C [12], as well as deposition of colloidal calcium phosphate in the micelle at temperatures above 100 °C. The irreversible formation of this less soluble calcium phosphate may also result in the formation of enlarged casein micelle aggregates if heated above 120 °C [13]. Immunogenicity as well as aggregation behavior changes when MP is heated in mixture compared to heating isolated whey proteins [14,15]. Therefore, heating of whey proteins in mixture with casein might result in different binding levels to sRAGE compared to their isolated forms. In this study, high-temperature heating was applied to MP powder in the presence of the milk sugar lactose at 130 °C for 10 min (HT-MP), as well as low-temperature heating at 60 °C for 3 weeks (LT-MP) to obtain a high level of the Maillard reaction with a high vs. limited level of protein aggregation. The samples were tested against an unheated control in sRAGE inhibition ELISA. Due to the high binding to the unheated control, further tests were conducted on micellar casein and sodium caseinate.

## 2. Materials and Methods

### 2.1. Chemicals

Soluble AGE product-specific receptor human *E. coli* (RD172116100) was obtained from Biovendor (Brno, Czech Republic). Anti-RAGE antibody (monoclonal mouse IgG2B clone, MAB11451) was purchased from R&D systems (Minneapolis, MN, USA). Horse radish peroxidase (HRP)-conjugated anti-mouse polyclonal goat (P0447) was purchased from Dako (Glostrup, Denmark). 3,3′,5,5′-Tetramethylbenzidine (TMB) substrate for high sensitivity ELISA was purchased from SDT reagents (Baesweiler, Germany). WesternBright^TM^ ECL Western blotting detection kit was obtained from Advansta (San Jose, CA, USA). Ovalbumin was purchased from InvivoGen (San Diego, CA, USA). Bovine serum albumin fraction V was obtained from Roche (Basel, Switzerland). Amyloid-β (1-42) ultrapure HFIP was purchased from Westburg (Leusden, The Netherlands). Raw bulk milk was obtained from the University Farm of Wageningen University (Wageningen, The Netherlands).

### 2.2. Methods

Figure 1 provides an overview of the experimental design.

#### 2.2.1. Preparation of Heated MP

Heating of MP was conducted as described by Zenker et al. [5]. Briefly, to obtain LT-MP, we heated powdered MP 60 °C for 3 weeks at a constant aw-level of 0.23. To obtain HT-MP, we heated powdered MP at 130 °C for 10 min (aw 0.23). Unheated powder was used as control (NT-MP). The powders were dissolved in simulated milk ultrafiltrate (SMUF) at pH 6.7, as described elsewhere [5]. SMUF was prepared according to the recipe of Jennes and Koops [16]. An aliquot of the dissolved NT-MP was acidified to pH 4.6 to deplete casein.

#### 2.2.2. Preparation of Micellar Casein and Sodium Caseinate

Micellar casein, as well as sodium caseinate, were isolated from raw milk as described by Moeckle et al. [17]. Briefly, micellar casein was isolated from raw milk by ultracentrifugation using an Avanti J-26 XP with JA-25.15 rotor (Beckman, Brea, CA, USA) at 65,000× *g* for 1 h. The pellet was washed with SMUF and centrifuged again. This step was repeated twice before freeze drying of the pelleted micellar casein. Sodium caseinate was obtained by acidification with sodium acetate buffer (pH 4.3). The precipitated sodium caseinate was centrifuged at 2000× *g* for 20 min at 4 °C and was washed twice with demineralized water. Afterwards, the sodium caseinate was freeze dried. The freeze-dried powders were dissolved in SMUF (pH 6.7) and centrifuged before analysis. Protein concentration was measured using Nanodrop ND200 (ThermoFisher Scientific, Waltham, MA, USA), according to the manufacturer’s instructions.

#### 2.2.3. sRAGE and CD36 Inhibition ELISA

Binding to sRAGE and CD36 was measured using inhibition ELISA as described by Teodorowicz et al. [8]. Briefly, MP samples before and after digestion were adjusted to 200 µg/mL protein concentration and incubated with sRAGE and CD36, respectively. Caseinate and micellar casein was diluted to different protein concentrations to measure a dilution curve ranging from 0.2 to 200 µg/mL. The dilutions were added to an ELISA plate blocked with glycated soy protein. Glycated soy protein was prepared by heating soy protein with glucose for 90 min at 100 °C in phosphate-buffered saline [2]. Afterwards, the plate was washed, and anti-RAGE antibody and anti-CD36 antibody, respectively, were added. Subsequently, anti-mouse polyclonal goat HRP conjugated antibody was added and detected using TMB substrate. Absorbance was measured using a Filter Max F5 multi-mode microplate reader (Molecular Devices, San Jose, CA, USA) at 450 nm vs. 620 nm reference wavelength.

Inhibition was calculated using the following formula:Inhibition [%] = (AbsMax − (AbsSample − AbsMin))/AbsMax × 100
where AbsMax is the absorbance obtained from the receptor without competition agent, AbsMin is the absorbance obtained from a blank sample (phosphate-buffered saline) without receptor, and AbsSample is the absorbance obtained from the mixture of receptor and each sample. High inhibition indicates high receptor binding affinity.

#### 2.2.4. sRAGE Dot Blot

sRAGE dot blot was performed for NT-MP, LT-MP, and HT-MP to assess differences in binding independent from their 3D structures. Analysis was conducted similar to the sRAGE Western blot described by Zenker et al. [2]. Briefly, 5 µg, 3 µg, and 1 µg of protein were spotted on a Amersham^TM^ Protran^TM^ 0.45 µm nitrocellulose membrane (GE Healthcare Life Science, Marlborough, MA, USA). The membranes were blocked with 3% bovine serum albumin in 1× Tris-buffered saline with 0.2% Tween (TBST) and incubated with sRAGE. Afterwards, anti-RAGE antibody was added, and the membranes were washed with TBST/Triton and TBST, before anti-mouse polyclonal goat HRP-conjugated antibody was added. ECL Western blot detection reagent was added for 30 s. Chemiluminescence was detected in ChemHighSensitivity mode using an Universal Hood III (Bio-Rad, Hercules, CA, USA) and Image Lab 4.1 software (Bio-Rad, Hercules, CA, USA).

#### 2.2.5. Simulated Infant In Vitro Digestion

Simulated infant in vitro digestion of NT-MP, LT-MP, and HT-MP was conducted following the protocol published by Ménard et al. [18]. Slight modifications were done as described by Zenker et al. [5]. Briefly, milk powder was dissolved in demineralized water to obtain 1.2% protein. Simulated gastric fluid was added to the samples, pH was adjusted to 5.3, and pepsin was added to a final activity of 268 U/mL digest. After 1 h of incubation at 37 °C, pH was adjusted to 6.6, and simulated intestinal fluid was added. Pancreatin was added to obtain a final trypsin activity of 16 U/mL digest, after which the sample was incubated for 1 h at 37 °C. Samples were taken after 60 min in the gastric phase (GP), 10 min in the intestinal phase (IP), and 60 min in the IP. After the GP, enzyme activity was stopped by increasing the pH to 6.6, while after the IP, enzyme activity was stopped using Pefabloc, as described by Ménard et al. [18].

#### 2.2.6. Statistical Analysis

Statistical analysis was performed with IBM SPSS version 25 using one-way ANOVA. Values were considered as statistically different at *p* < 0.05.

## 3. Results and Discussion

In the dairy industry, heating of MP is an essential tool to ensure product safety and to obtain powdered products. However, this may also affect the allergenicity and immunogenicity of the products [14,19]. Inhibition ELISA for AGE receptors can be used as a screening tool to indicate whether a specifically treated protein is a potential ligand for AGE receptors and thus could initiate an immune response. While whey proteins have been extensively studied regarding their binding to especially the receptor for AGEs (RAGE), products that also contain caseins have not been investigated. NT-MP, LT-MP, and HT-MP showed binding (>50% inhibition) to sRAGE in the inhibition ELISA (Figure 2).

Interestingly, LT-MP showed 10.8 percent-point lower binding than NT-MP, whereas HT-MP did not show significantly different binding from NT-MP (Figure 2a). sRAGE is known to bind to protein-bound AGEs such as N^ɛ^-carboxymethyl lysine (CML) and N^ɛ^-carboxyethyl lysine (CEL) [3,4,7]. The conditions used to make LT-MP and HT-MP samples have previously been shown to lead to significant higher levels of CML, CEL, and pentosidine with increasing heating temperature [5]. Furthermore, the binding of sRAGE to β-lactoglobulin-bound CML has been reported to depend on the CML modification level [3]. However, this was not confirmed for our MP in mixture samples. It was hypothesized that casein might interfere in the inhibition ELISA, resulting in the high binding observed for the MP samples independent from the heat treatment. Therefore, casein was depleted in NT-MP and measured with sRAGE inhibition ELISA (Figure 2a). The high binding of sRAGE to NT-MP, accompanied by the diminished binding to casein-depleted NT-MP (Figure 2a), indicates that there is an interference of casein in the inhibition ELISA, which could explain the similar binding of NT-MP and the heated samples. This was confirmed by the binding of unheated micellar casein and sodium caseinate to sRAGE. They both showed binding starting at 20 µg protein/mL, which increased with higher protein concentrations (Figure 2b). The same effect of casein and caseinate was also observed in CD36 inhibition ELISA (Figure 2c), indicating that casein also shows interference with AGE receptors of the scavenger family. This demonstrates that both sRAGE and CD36 bind to casein, independent of heat-induced modification. Therefore, the presence of casein can interfere in ELISA assays when determining binding of AGE receptors to heated MP. Next to AGEs, it has also been shown that other 3D-structural properties such as hydrophobicity and aggregation of food proteins can result in binding to AGE receptors [2,3,8]. Therefore, the role of the 3D structure of casein in binding to sRAGE was further investigated using dot blot analysis (Figure 3).

When using dot blot, we observed binding of sRAGE to HT-MP but not to NT-MP and LT-MP at all protein concentrations (Figure 3). This is in contrast to the findings from ELISA, where binding was also observed to NT-MP and LT-MP (Figure 2). Notably, a previous study showed that most of the casein becomes insoluble in the heated samples [5]. However, the binding observed in the dot blot is in line with the higher level of AGEs observed in HT-MP, which have previously been measured for the MP samples in another study [5]. This indicates that there is indeed an effect of heating and glycation on binding to sRAGE for this sample that could not be observed in ELISA due to the interference of casein. The absence of sRAGE binding to NT-MP in the dot blot (Figure 3) as well as the binding to micellar casein and sodium caseinate (Figure 2) shows that the binding of sRAGE to this sample that was observed in ELISA (Figure 2a) is dependent on the presence of casein particles in the sample solutions. Both sRAGE and CD36 are promiscuous receptors that bind various ligands [9,20]. For both receptors, negative charge, aggregation, and hydrophobicity have been suggested as important determinants for their binding to ligands [3,7]. Moreover, there is evidence in literature that, e.g., sRAGE recognizes high molecular weight ligands formed upon aggregation of proteins [2,8,10]. The hydrophobic elements of the casein molecules are located in the inside of the casein micelle, whereas the negatively charged ends of the κ-casein point to the outside of the micelle [11]. However, binding was also observed to caseinate, which does not show this particular micellar structure. It is thus hypothesized that one of the main drivers for the binding to casein is the size of the casein micelle, which has been reported to range between 50 and 100 nm radius and a molecular weight > 3.7 × 10^8^ Da [11,21]. However, also binding of sRAGE to sodium caseinate appears to be related to size, as it is known to contain clustered casein molecules that have a smaller hydrodynamic radius (≈10 nm) compared to micellar casein and a molecular weight of 200–250 kDa [22]. Nevertheless, the absence of binding to NT-MP and LT-MP in the dot blot indicates that the 3D structure of the casein micelles and caseinate is responsible for the binding observed in ELISA. The binding of proteins to nitrocellulose membranes predominantly occurs due to hydrophobic interactions. However, the casein micelle consists of a hydrophobic core, where the hydrophilic, glycosylated ends of κ-casein reach to the outside of casein [11]. It is known from milk homogenization that casein micelles partly spread over the fat globule surface and cover the newly formed oil–water interface probably via hydrophobic interactions between the caseins and the oil phase [23]. Upon this process, they partly lose their micellar structure, and it could be assumed that such a process also happens upon binding of micellar casein to the hydrophobic nitrocellulose membrane. This probably explains why the structural effect on the binding of casein observed in the ELISA are not reflected in the dot blot.

There is a rising interest in including in vitro digestion experiments in the evaluation of immunogenicity and allergenicity of food proteins [24]. This has already been performed in some studies that investigated immunogenicity of MP [8,25]. As casein is hydrolyzed upon digestion, it was verified whether the interference of casein in inhibition ELISA for sRAGE and CD36 binding is still observed after in vitro digestion at different time points of digestion. In an adult in vitro digestion model, the pH in the GP is below the isoelectric point of casein (pH 3.0) and thus results in its depletion, which, as shown in Figure 2, diminished the interference of casein in sRAGE ELISA. In an infant in vitro model, where the pH in the GP is higher (pH 5.3), and the resulting depletion of casein is therefore less, the binding of sRAGE and CD36 to NT-MP however remained (Figure 4).

Next to this, no differences were observed in the GP between samples for sRAGE, while CD36 showed lower binding to LT-MP and higher binding to HT-MP, compared to NT-MP. However, differences were only minor. In the IP, the binding of both receptors to all samples strongly decreased. This is probably related to the rapid enzymatic hydrolysis of casein in the IP. Moreover, also sRAGE and CD36 ligands that are formed in LT-MP and HT-MP upon heating and glycation may be hydrolyzed, explaining the low binding to HT-MP after intestinal digestion.

To summarize, this study demonstrated that the natural structure of casein as well as caseinate clusters induces binding to sRAGE and CD36 and interferes in the assessment of the formation of AGE-receptor ligands in heated milk using ELISA. Moreover, these data show that after in vitro digestion in the IP, interference of casein in sRAGE and CD36 inhibition ELISA diminishes due to the hydrolysis of the MP. Despite the interference of casein in the ELISA, dot blot analysis might present a solution to monitor the heat-induced changes in receptor binding of casein prior to digestion. It could be hypothesized that the interference of casein may also occur in other cell assays if the results are based on receptor mediated mechanisms. Van Der Lugt et al. [25] previously investigated the effect of glycation of casein in THP-1 cells and found no secretion of tumor necrosis factor-α after exposure to unheated casein. This indicates that casein does not automatically also interfere in a cell-based assay. However, this likely depends on the cell type, the investigated parameter, and the receptors that are expressed on the cells.

## 4. Conclusions

We conclude that sRAGE and CD36 binds to casein, independent from the application of heat treatment, and thus the extent of glycation, which should be considered when testing cow’s MP in mixture. The binding to these receptors decreased after intestinal digestion of casein. Moreover, it is suggested that interference of casein micelles, when performing other analyses on receptor binding or receptor-mediated reactions of immune cells towards heated MP, should be taken into consideration. Our results indicate that binding of RAGE can be the result of multiple factors that can be found in foods, and not just of AGEs.

## Figures and Tables

**Figure 1 foods-10-01836-f001:**
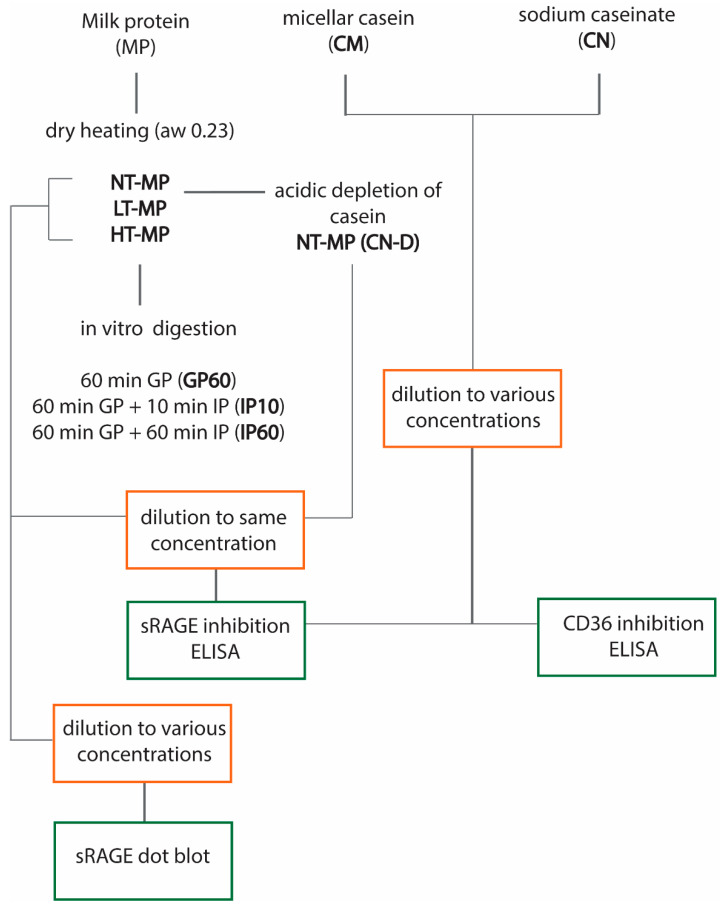
Overview of the experimental design. Non-treated milk protein (NT-MP), low-temperature-heated milk protein (LT-MP), high-temperature-heated milk protein (HT-MP), casein depleted (CN-DP), gastric phase (GP), intestinal phase (IP), water activity (aw), and soluble receptor for advanced glycation end products (sRAGE).

**Figure 2 foods-10-01836-f002:**
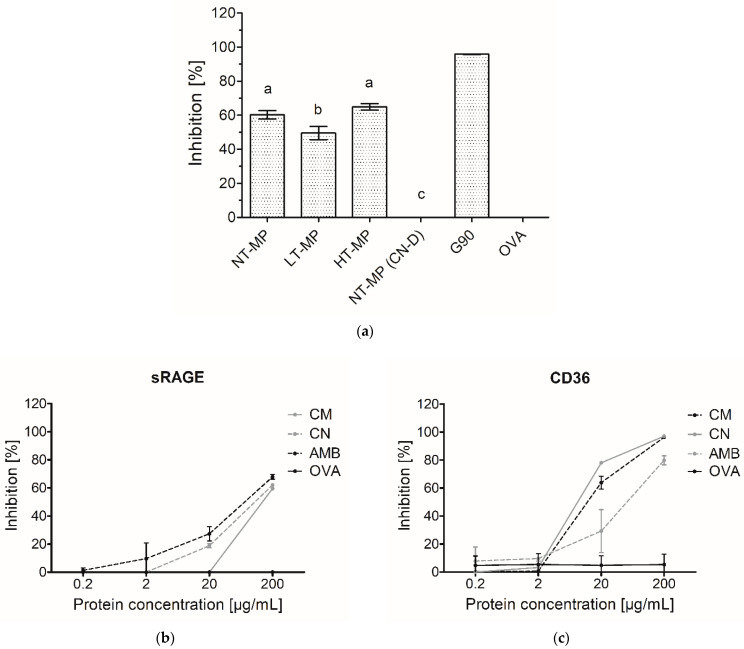
(**a**) sRAGE inhibition ELISA on 200 µg/mL cow’s milk protein (MP), non-treated (NT-MP), low-temperature-heated (LT-MP), high-temperature-heated (HT-MP), and NT-MP after acidic depletion of casein (CN-D). (**b**) sRAGE inhibition ELISA and (**c**) CD36 inhibition ELISA on micellar casein (CM) as well as sodium caseinate (CN) measured at different concentrations. Amyloid-β (AMB) and soy protein glycated with glucose for 90 min at 100 °C (G90) were used as positive controls, while ovalbumin (OVA) was used as a negative control. Statistical differences in (**a**) were determined using one-way ANOVA. Values were considered as statistically different at *p* < 0.05, with different letters indicating statistically significant differences between samples.

**Figure 3 foods-10-01836-f003:**
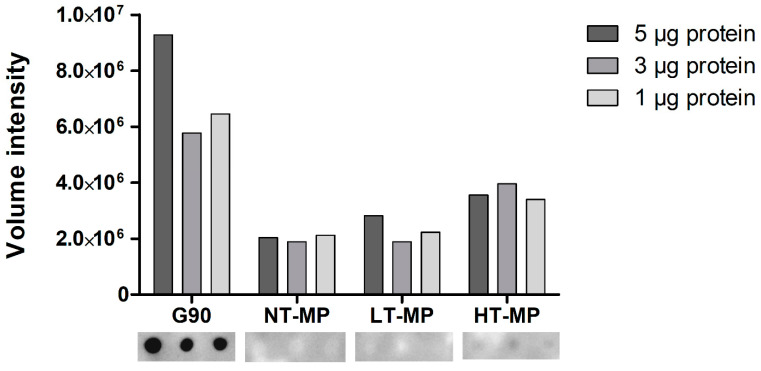
sRAGE dot blot volume intensities (top) and dot blot images (bottom). Cow’s milk protein (MP). MP was non-treated (NT-MP), heated at low temperature (LT-MP), and heated at high temperature (HT-MP), while soy protein, glycated with glucose for 90 min at 100 °C (G90), was used as positive control.

**Figure 4 foods-10-01836-f004:**
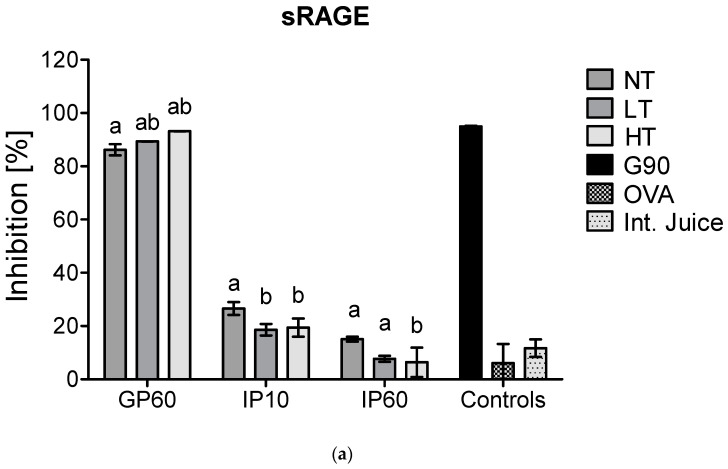
sRAGE inhibition ELISA (**a**) and CD36 inhibition ELISA (**b**) of simulated infant in vitro digests of 200 µg/mL milk protein, non-treated (NT), heated at low temperature (LT), heated at high temperature (HT), and the intestinal juice (Int. Juice). Soy protein, glycated with glucose for 90 min at 100 °C (G90), was used as a positive control, while ovalbumin (OVA) was used as negative control. Statistical differences within the same digestion time point were determined using one-way ANOVA. Values are considered as statistically different at *p* < 0.05, with different letters indicating statistically significant differences between samples.

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
