# Peer review of "No Glycation Required: Interference of Casein in AGE Receptor Binding Tests"

_foods, 2021, doi:10.3390/foods10081836_

Round 1

Reviewer 1 Report

Line 36: more details of milk protein fractions can improve this part. For example caseins-whey proteins proportions and allergenicity.

Line 87: Nanodrop: a suitable reference and methods shoud be added.

Line 244: more results can be added to the conclusion part.

Author Response

We would like to thank the reviewer for the useful comments on our manuscript. Below, are our detailed response to the issues raised.

Line 36: we have added additional information here on both the ratio and the allergenicity of the two categories of milk proteins (line 35-39).

Line 87: Details have been added (line 89).

Line 244: As we wrote this as a “Brief Report” we tried to keep the manuscript short. In the revised version, we added some more results to the conclusion (line 262-270).

Reviewer 2 Report

The authors report a study focused on the effect of heat treatment on the immunogenicity and allergenicity of cow’s milk proteins. The work is based on the recognition of the soluble form of the receptor for AGEs (sRAGE) toward milk proteins, which can mediate the immunogenic response. In particular, the research aims to investigate the role of caseins in milk proteins sRAGE’s binding, upon heat treatment at high and low temperatures. The main goal of the work is to assess the effect of different degrees of heat treatment on the binding of milk proteins to sRAGE. This is an interesting topic that can have an impact on the effect of milk processing on cow’s milk protein allergy (CMPA) in infants.

The methodological approach is mainly based on the sRAGE inhibition ELISA assay, but the study design remains very vague, preventing the reader to get an easy comprehension of the meaning of the experiments.

I would recommend a more clear description of the experimental design; it would be helpful to add a figure or a scheme with all the samples tested.

This lack of clarity is even more striking in the Result session: the information is given without a clear order, and the way in which is commented is sometimes confusing.

I suggest the authors do a deep revision of the paper, in order to make clear the meaning of the results obtained.

Main criticisms:

- Figure 1. a:  it is not clear which is the concentration of the protein samples under analysis (“same protein concentration”, reported in the methods, is not quantitative information..)

- Which quantitative parameters can be derived by the inhibition ELISA assays?

The percentage values are only in one case reported and without a standard deviation.

- The data are not few, but not well described and the conclusion remains very vague.

I think that the effect of for all the conditions tested in which temperature could have an impact (formation of NÉ›-carboxymethyl lysine (CML), NÉ›-carboxyethyl lysine (CEL),  3D structure, caseins micellisation) should be better rationalized.

Author Response

We would like to thank the reviewer for the useful comments on our manuscript, which we used for revising the manuscript as detailed in our response to the issues raised:

-To avoid confusion about the experimental design, we have added a figure with an overview of this approach (Fig 1). We think that this also helps to understand the results better.

-We have added better explanations and rephrased sentences to improve clarity as well as readability: Lines 144-151, 160-164, 167-170, 180-182, 195-203, 225-227

-Information on protein concentration has been added to materials & methods (line 96-99) and figure legends.

-The ELISA used is suitable to compare among samples but does not give quantitative results about the amount of ligand bound to the receptor. It is commonly used as an initial screening to determine whether a protein is a ligand for the investigated receptors. Additional explanation was given in 144-151. Standard deviations are given for all samples in all assays (except for the dot blot); in some cases they are however very small.

-Regarding the rationalization of the overall conclusion, we do not have a full explanation for all results. Still, we believe that the results are very useful for others working in this field and therefore decided to write this as a brief report. We have now added a sentence to the conclusion indicating that multiple causes may underlie the results found, but that at least it is clear that more than only AGEs are playing a role (line 267-269).

Round 2

Reviewer 2 Report

Dear Editor,

I think the authors have addressed all the criticisms I had pointed out.

Thanks

Author Response

Thank you for checking our manuscript and the feedback given.